# Different Production Processes for Thermoplastic Composite Materials: Sustainability versus Mechanical Properties and Processes Parameter

**DOI:** 10.3390/polym15010242

**Published:** 2023-01-03

**Authors:** Marco Valente, Ilaria Rossitti, Matteo Sambucci

**Affiliations:** 1Department of Chemical Engineering, Materials, Environment, Sapienza University of Rome, 00184 Rome, Italy; 2INSTM Reference Laboratory for Engineering of Surface Treatments, UdR Rome, Sapienza University of Rome, 00184 Rome, Italy

**Keywords:** thermoplastic matrix composites, impregnation processes, manufacturing, fiber reinforced, processing

## Abstract

Up to now, fiber-reinforced composites with thermoplastic matrix have seen limited fields of use in the structural scope due to their high viscosity in the molten state, which results in poor impregnability of the reinforcement, leading to mechanical properties of the finished product that are not comparable to those of thermosets. Although the latter still dominate the various sectors of automotive, aerospace, transportation and construction, new applications involving the production of thermoplastic composites are growing rapidly, offering new approaches to the solution of this problem. The aim of this work is to study and evaluate the state of the art on the manufacturing processes of thermoplastic matrix composite, analyzing the parameters that come into play and that most influence the process and material performance. The advantages of film stacking and powder impregnation techniques are contrasted by the versatility of hybrid fabrics and, at the same time, parameters such as pressure and temperature must be carefully considered. A description of different thermoplastic composite processes such as powder impregnation, film stacking molding, hybrid woven fabric, hybrid yarn and products follows, which represent the current possibilities to move from a thermosetting matrix composite to a thermoplastic one, upon which the concept of sustainability is based. This article wants to present an overview of research that has been done in manufacturing thermoplastic reinforced composites and will serve as a baseline and aid for further research and development efforts.

## 1. Introduction

Thermoplastic composites (TPCs) offer some advantages over thermosets, such as higher toughness, faster production and, first of all, their recyclable nature [1]. Many novel techniques have been proposed, developed and evaluated for thermoplastic applications in the last 20 years and, in recent years, continuous fiber reinforced thermoplastic matrix composites have been successfully employed in the aircraft, military and aerospace industries due to their excellent properties [2]. In these and many other commercial engineering applications, they can replace other materials, such as thermoset composites [3]. Since the late 1990s, an ever-increasing number of aerospace products in these materials have moved into series production. They allow a completely new and automated production since they can be heated and formed repeatedly [4]. They have several advantages over other materials, including excellent fire resistance, significant strain-to-failure, better impact tolerance, higher resistance to fatigue, short molding time and longer shelf life of prepreg [5]. However, the high cost and difficulties of impregnating continuous fiber thermoplastic composites, resulting from polymer melting or the use of solvents, still limits their use in commercial applications. Therefore, cost savings and mechanical properties largely depend on developing more efficient methods for impregnating fibers due to the thermoplastic’s high viscosity and processing the final composite parts [3]. Furthermore, their use has been limited by the lack of good processing techniques to make flexible prepreg tapes. The general intractability of these matrices due to their high viscosity and high processing temperatures gives rise to a series of problems, such as the manufacture of stiff tape, devolatilization of the solution which leads to the formation of voids with the consequent loss of mechanical properties, and dishomogeneous resin content prepreg tapes [6]. A solution to improve fiber impregnation is to bring the matrix and fibers into contact as much as possible before the final molding step or, in other words, reducing the flux length required by the polymeric matrix. Various concepts of these intermediates have been developed [1], such as commingled textiles that consist of both reinforcing and polymer fibers, textiles made of powder coated fibers and partially or fully consolidated panels. As Figure 1 shows, in the general context of thermoplastic polymer matrix composite manufacturing, two approaches can be taken: the most direct, Reactive Processing, which starts from the fabric impregnation with the monomer and involves in-situ polymerization. We will not discuss this process in this paper as it has been addressed and explored in detail in another work [7] by our research group. Instead, our attention will be focused, in this article, on the second technique, Melt Processing. It is important to emphasize the fact that in-situ polymerization can be the best performing route as it might be able to limit as much as possible the porosity and difficulties arising from Melt Processing, e.g., the inability to achieve optimal impregnation due to high viscosities of thermoplastic melt material. This, in turn, could lead to lower mechanical properties and a fair amount of defectiveness, which needs to be minimized. While Reactive Processing is better performing, the monomers that can fulfill this function with feasible and economically viable industrial processes are limited. To date, it is only possible with PA6, starting with ε-caprolactam monomer and PMMA. With Melt Processing, therefore, it is more difficult to get a defect-free composite, but it is possible to use a multitude of polymers and thus dose the properties of the final product.

Nevertheless, one of the most effective solutions lies precisely in their production, in order to move toward eco-sustainable and circular composites. The strong commitment of world organizations in the field of safeguarding the planet has directed the research of these materials toward production processes with a lower environmental impact and a strong propensity to recycle the polymeric part [8]. The development of eco-sustainable composites has involved the study of glass/polymer [9], carbon/polymer [10], metal/polymer [11], polymer/polymer [12], ceramic/polymer [13], and concrete/polymer [14] interactions for the optimization of the final mechanical properties and the inclusion of these products in the perspective of environmental and economic circularity.

The rapid increase in the application of fiber-reinforced polymer matrix composites is creating a challenge for waste recycling. The use of high-performance thermosetting polymers as a matrix makes the recovery of fibers and resins extremely difficult. Implementing a circular economy able to eliminate waste and reuse resources ensures the use of efficient processes to recycle thermoset composites components and manufacturing wastes [15].

Definitely, the outstanding performance of conventional thermosets arising from their covalently cross-linked networks directly results in a limited recyclability. The available commercial or close-to-commercial techniques facing this challenge can be divided into mechanical, thermal and chemical processing. However, these methods typically require a high energy input and do not take the recycling of thermoset matrix itself into account. Rather, they focus on retrieving the more valuable fibers, fillers, or substrates [16]. This push in the use of thermosetting composites is now raising awareness of their fate: a direct consequence is, in fact, a strong increase in the relative waste, coming from the production processes (prepreg waste; cuttings and scraps of vulcanized composites, which represent approximately 30–40% by weight of total materials) and, belatedly, by end-of-life products (EoL). In fact, carbon fiber reinforced polymers worldwide are projected to reach up to 20 ktons per year by 2025 [17]. With approximately 6000–8000 commercial aircrafts reaching their end-of-life by 2030 [18], there is a clear and demanding request in order to develop economically-sustainable waste management and recycling techniques for fiber reinforced polymer matrix [15]. As shown in Figure 2, the global epoxy composite market size was valued at USD 28.40 billion in 2020 and is expected to grow at a compound annual growth rate (CAGR) of 8.3% from 2021 to 2028 [19]. Rapid replacement of conventional materials in automotive and aerospace applications is expected to drive the market in the coming years. Furthermore, it should also be emphasized that the current European Union (EU) legislation still lacks specific regulation for the treatment of composite waste. Some hints are included in the 2000/53/EC EU Directive, which provides for a recovery of 95% and a recycling rate of 85% of the total weight of the EoL vehicle and limits the use of non-metallic components if they do not comply with the requirements of the Directive, but no specific instruction on how to treat EoL carbon fiber reinforced polymers is specifically addressed [15] or these reasons, industry and researchers have shown a growing interest in new approaches for the recycling of thermosetting composites or, in the most innovative solutions, on the potential of manufacturing thermoplastic composites.

It is a fact that, when taking into account factors such as climate change, global warming, environmental sustainability and circular economy, the landfill or incineration of thermoset component wastes must be avoided. In this context, more efforts are required to improve the technology readiness level of the processes in place and their scalability should be economically accessed. To develop commercially viable recycling activities, the future research studies must be focused on the following points: achievement of consistent quality of recycled fibers; reuse of them as reinforcement in thermoplastic polymers, also from renewable sources; study of mechanical properties after reuse and remanufactured technologies evaluation of the potential to close the life cycle loop of CFRPs and reducing energy consumption and recycling cost. In this field, innovation is multifaceted and can come both from the constituent materials (high-performance engineering thermoplastic matrices, commingled semi-preg of recycled carbon, and glass fibers/thermoplastics) and production processes (film stacking, dry powder impregnation, fiber commingling).

This review will be focused on the latest advances and development of reinforced thermoplastic, mainly referring to continuous fibers and non-woven fabric composites and related processing technologies: relationships between impregnation mechanisms, consolidation quality and resulting mechanical properties of composites manufactured from film stacking, commingles fabrics and dry powder impregnation systems will be investigated. This paper highlights how these materials can be processed, adopting a completely different paradigm in comparison to that of the thermosetting matrix, the former not requiring a reaction to set the properties but complex thermal cycles, often performed in a short time span.

## 2. Impregnation Methods from Molten Polymer Matrices

It is possible to make an important distinction between various impregnation methods: we distinguish those that precede it, called pre-impregnation, and those that follow it, called post-impregnation. In the former, the fibers impregnation takes place in a single step using a polymer in the molten state. The various layers thus obtained are stacked and subsequently consolidated due to the application of heat and pressure. In the latter, however, it is expected that the impregnation occurs during the processing of the piece starting from a polymer in the shape of a film, powder or filament [20]. In this article we will focus only on the post-impregnation phase starting from semi-impregnated forms.

Figure 3 presents several methods for wetting the fibers with a thermoplastic polymer. Different methods have been employed to make thermoplastic pre-impregnated tapes. The most common processes for combining the polymer and the reinforcement that will be addressed in this work are:Film stacking, technique that consists in alternating thin polymer sheets with the reinforcement and, subsequently, compacting them;Powder impregnation technique consists in covering and, therefore, impregnating the fiber fabric with the polymeric matrix in the shape of a dry powder [21];Hybrid woven fabrics;Hybrid yarns.

### 2.1. Film Stacking

Film stacking consists in heating and compressing a sequence of alternated layers of thermoplastic films and dry fabric reinforcements (Figure 4). The resulting thermoplastic laminates serve as a semi-finished product subsequently intended to thermoforming processes to obtain the desired shape of the final composite component. Film stacking process involves three main phases [22]:Heating of the press to lower the matrix viscosity;Increasing of the pressure to force the liquid-state thermoplastic matrix to impregnate the fabric;Cooling of the press to solidify the laminate.

Temperature, pressure, and holding time are the core parameters to control the process. Their values depend on the nature of the polymer, mainly melting temperature and viscosity, as well as on the fabric tortuosity and the load and temperature the fibers can sustain [23]. Below a comprehensive literature survey on the influence of these process parameters on the properties of film-stacked thermoplastic composites is reported.

#### 2.1.1. Influence Mechanism of Processing Temperature

Processing temperature considerably affects the mechanical and microstructural properties of the composite produced. It is known that the use of thermoplastic resins introduces the issue of adequate fiber bed impregnation because of the higher viscosity of thermoplastic melts than thermosetting ones. Optimization in processing temperature is crucial. High temperature pressing scheme promote a lower resin’s viscosity and therefore greater mobility and fluidity to penetrate between the fibers and impregnate them. On the other hand, the use of high temperatures potentially clashes with the thermal stability of the materials constituting the composite. Then, an optimal condition will exist where adequate resin flow occurs during consolidation (i.e., good impregnation and composite quality) and thermal degradation of the constituents is kept at a minimum [24].

Hu et al. [25] explored the influence of molding temperature (360–400 °C) on the mechanical strength of carbon fiber fabric reinforced polyether ether ketone (CFF-PEEK) thermoplastic composites. The tensile and flexural properties exhibited the char of first increase and then decrease in the temperature range under investigation. From 360 to 390 °C the tensile and flexural strengths increased from 412.63 MPa and 487.45 MPa (+15.35%) and from 663.21 to 808.32 MPa (+17.95%), respectively. At 400 °C, a negative influence in mechanical properties was detected, with a decrease in tensile and flexural strengths of 9.95% and 11.88%, respectively. The authors explained the results monitoring the viscosity of PEEK melt under different isothermal conditions. At the molding temperature of 390 °C, the resin showed the lower viscosity and the higher fluidity, adequately impregnating the fibers and providing the composite the best performance. At 400 °C, PEEK converged to a rapid increase in viscosity because to thermal crosslinking of macromolecular chains induced by excessively higher temperature. Consequently, the increment in viscosity hindered the proper impregnation of carbon fabric, worsening the mechanical performance. Kim and Park [26] investigated the effect of process temperature (205, 215, and 225 °C) on the impregnation quality of flax fiber reinforced polypropylene (FFR-PP) composites. The authors discovered that, at the lowest temperature, the flax fabric was not completely impregnated. Although the higher temperatures are advantageous with respect to the resin flow, the void content in the laminate fabricated at 225 °C (5.1%) was greater than the process temperature of 215 °C (3.2%). In this case, the thermal degradation of flax fiber induced by the highest processing temperature was recognize as the main reason for the increment of void concentration at the matrix-fiber interface. Then, detrimental effects were detected in terms of flexural properties. Zal et al. [27] studied the effective processing temperature (160–240 °C) on the strength and microstructural properties of glass fiber reinforced polyvinyl chloride (GF-PVC) composites. Due to the reduction of PVC viscosity, impregnation and flexural strength of the samples get better by increasing the processing temperature up to 220 °C (from 59.4 MPa at 160 °C to 146 MPa at 220 °C). The highest temperature (240 °C) involved degradation of PVC matrix, weaking the bonding between fibers and matrix and dramatically diminishing the flexural performance (39.3 MPa). Figure 5 elucidates the different failure mechanism as a function of processing temperature (at the same pressure and holding time). At the lowest processing temperature (160 °C) the impregnation quality was low, causing inter-laminar sliding failure (Figure 5a). By increasing the processing temperature, the adhesion between matrix and GF gradually improved. At 180 °C, both inter-laminar sliding and fibers fracture occur simultaneously like failure mechanism of the composite (Figure 5b). For the laminate produced at 240 °C, the failure mode was governed by the degradation due to the weak adhesiveness and integrity of matrix (Figure 5c).

#### 2.1.2. Influence Mechanism of Pressure

The pressure drives the thermoplastic resin to infiltrate into the compliant fabric while simultaneously deforming it, altering the infiltration kinetics [28]. This parameter plays counteracting roles to determine the resin impregnation in the fibrous medium. Too low compression pressure can increase the impregnation time, forming dry spots inside the composite. Conversely, if too high pressure is applied on the fibrous bed, the fiber volume fraction increases decreasing the fabric’s permeability due to the reduction of the pores volume between the bundles and between the fibers [23,26]. Additionally, the pressure must be carefully selected to avoid excessive compression loads exerted on the fibers, which lead to damage or failure to the reinforcement prior the composite’s fabrication [29].

Suresh and Kumar [30] researched the effect of forming pressure (4 MPa, 7 MPa, and 9 MPa) on the mechanical properties of glass fiber reinforced polypropylene (GF-PP) laminates. Keeping fixed the processing temperature at 190 °C, the authors observed that an increase in the forming pressure initially increased the flexural and tensile strengths (+17% in tensile and +63% in bending, moving from 4 MPa to 7 MPa of pressure), then decreased the properties of the composite laminates (−12% in tensile and −30% in bending, moving from 7 MPa to 9 MPa of pressure). The optimal pressure condition of 7 MPa, also corroborated by Design of Experiment (DOE) computational analysis, revealed an improved wetting of the glass fibers and therewith good bonding to the matrix in the GF-PP system. Katayama et al. [31] optimized a film stacking process to fabricate jute fabric reinforced poly-lactic acid (JF-PLA) composites. Investigating among a range of compaction pressures (1 MPa, 2 MPa, 3 MPa, and 4 MPa) at 160 °C as a processing temperature, the authors discovered that that the higher moulding pressure than 3 MPa did not significantly affect the impregnation and density of composite structure. Moulding pressure of 3 MPa was consider optimum in terms of moulding time (300 s), microstructural compaction, and mechanical strength reaching approximately 115 MPa in flexural strength. Kazano et al. [32] investigated the relation among mould pressure conditions in film-stacking process and resin impregnation to fiber yarns in carbon fiber fabric reinforced polyimide (CFF-PI) composites. The impregnation ratio, predicted by the Kobayashi et al. [33] model, and the microstructural analysis of the laminate were assessed as a function of five pressure conditions (0 MPa, 1 MPa, 2 MPa, 4 MPa, 8 MPa). The photograph in Figure 6 highlights that since the polymer impregnation was not improved after the pressure 4 MPa, this condition was indicated as the optimum molding pressure. Interestingly, it is possible to verified different void size and distribution as a function of processing pressure. At 0 MPa (no compaction), a large void can be detected at the center of the yarn (Figure 6a). By increasing the pressure, fine voids are formed inside the fabric and their dimensions tend to gradually diminish. This is because the pressure disassembles the large-sized gap in the uncompacted laminate to the fine voids, which tend to disperse in the composite. The increase in mould pressure also promotes the expulsion of excess air resulting in smaller void sites. Consistently, at the optimum pressure of 4 MPa, the impregnation ratio converged to 100%.

#### 2.1.3. Influence Mechanism of Holding Time

Other than temperature and pressure, the application time for these parameters is another critical consideration in the film stacking process. Insufficient holding time could compromise the laminate’s microstructural quality, producing defects on the sample, which are associated with inadequate pressure and temperature [34].

As with processing pressure, Katayama et al. [31] investigated the influence of different holding times (30–600 s) on the impregnation of JF-PLA laminates. Cross section photos of the film-stacked composites processed at 1 MPa and 160 °C revealed that the high melt viscosity of PLA as well as the twist architecture of natural fiber bundles required almost 300 s for an adequate and defect-free impregnation. After 300 s the polymer impregnated into fiber bundles but no significant improvement in flexural strength occurred, verifying that a “threshold” duration for compression exists which provides short working time and good mechanical performance, see Figure 7.

Yang et al. [35] analyzed the influence of holding time on the mechanical behavior of ultra-thin carbon fiber fabric reinforced polycarbonate (CFF-PC) composites. The examined moulding times ranged from 0.5 min to 5 min at increments of 1.5 min, considering a processing temperature of 240 °C and pressure of 6 MPa. Although, long holding time would seem to promote the wettability and interfacial bonding between the matrix and reinforcement, maximum tensile properties were identified at 3.5 min (tensile strength of approximately 450 MPa) while the strength decreased when the parameter reached the higher value of 5 min. The authors explained the detected results by considering the excessive increase in flowability of PC matrix for the highest process parameters (temperature, pressure, and holding time). The resulting rheology of the melt polymer would generate an extremely high scouring force, disturbing the carbon fiber location and the pre-determined uniformity. Then, misalignment of the reinforcement would act as a manufacturing defect, negatively altering the strength of the laminates. Vitiello et al. [36] implemented two impregnation procedures, in terms of duration and intensity, for the fabrication of basalt fiber-reinforced polyamide 11 (BF-PA) laminates. Regarding “slow” procedure the laminate was compacted at lower pressure (3 MPa) and for longer times (13 min), suffering from the growing viscosity of the matrix. The “fast” procedure applied high pressure (4.5 MPa) at short time (7 min), that is, when the polymer viscosity is low. The differences in fabric impregnation resulted in substantial differences in the mechanical properties of the laminates: “fast” procedure sample exhibited better fiber impregnation, lower porosity, higher flexural modulus (+20%) and strength (+60%), and even the damages following impact and indentation stresses were lower than those recorded in the sample realized with the slow procedure.

### 2.2. Powder Impregnated Tow

The use of thermoplastic composites has been limited by the lack of reliable techniques to make prepregs of consistent quality. Dry powder impregnation methods offer a new approach to the solution of this problem. This process was identified as the technique with the greatest potential for success as a valid alternative to produce prepreg thermoplastic tapes [6]. Various research groups have developed several ways to disperse polymer powder and impregnate a fiber tow as summarized in Table 1.

These processes differ mainly in the way in which the particles of the matrix are deposited on the fibers and in the forces of particle/fiber interaction responsible for their adhesion. In most dry processes, the fibers are passed through an air suspension of particles. Some of the main factors that influence the mode and rate of deposition of the powder are:Fiber tow spreading;Particulate flow pattern;Particle size;Dielectric properties of fibers and particles;Particle cloud concentration [21,37].

It is expected that the powder is introduced into a fiber tow which is processed by heating to sinter the particles onto the fibers. Price [38] was the first to employ this technique, that provides for the passage of the glass roving through a bed (fluidized or loosely packaged) of thermoplastic powder. The particles attach themselves to the fibers due to electrostatic attraction. The tow is then heated and passed through a mold to produce an impregnated tow. The impregnation is macroscopic, i.e., the particles cover fiber groups rather than single fibers and is mainly aimed at the production of short fiber reinforced thermoplastics. Polypropylene particles with an average diameter of 250 microns were used. Ganga [6] fluidized polyamide particles smaller than 20 microns in a fluidization chamber, impregnated glass rovings and then coated it with an outer sheath of a second material with a lower melting point than the impregnated particles. The second sheath was extruded onto the tow. Muzzy et al. [39] demonstrated the ability to manufacture the prepreg by passing a scattered tow through an electrostatic fluid bed of PEEK powder (50 microns). Allen et al. [40] impregnated the fibers in a recirculation chamber with annular walls to aid dust recovery and a fan on the bottom to disperse the thermo-plastic polyimide particles (5–20 microns, LaRC-TPI) [6].

Powder polymer matrix and reinforcement can contact each other electrostatically. The impregnation takes place thanks to the charged powder particles deposited on the fabric surface [20,39,41]. In this case as well, the consolidation occurs through the simultaneous application of heat and pressure in order to obtain continuous fiber prepregs [6,41]. Rath et al. [42] produced prepregs with a Nylon-12 matrix and a reinforcing volumetric content between 20% and 50% in continuous aramid fibers [20]. Padaki and Drzal [37,43] identified heating, compression and cooling as the three necessary steps for the process. This method is discussed in detail in literature.

The flowchart for the manufacture of powder-impregnated thermoplastic composites is shown in Figure 8 and lists the two basic steps:Tape manufacture, which involves the fiber tows impregnation with thermoplastic particles and their subsequent coalescence on the fibers to form a flexible prepreg tape;Consolidation, which involves laying prepreg tapes into a mold followed by heat and pressure application to form void-free composites. An important aspect for optimizing the processing cycle is the characterization of the different properties of the thermoplastic matrix and the reinforcing fiber.

Dry powder processes have numerous advantages for the production of thermoplastic prepreg tape. An ideal process would have the following advantages over other processes:It would be little affected of the viscosity of the matrix. Most high-performance thermoplastic matrices are highly viscous (104 to 105 Poise) above their softening point (amorphous) or melting point (semi-crystalline). A good dry powder process would circumvent this problem by coating the fibers individually so that flow occurs over very short distances of the order of microns;It would avoid the use of binders, solvents or water that must be evaporated during the last stages of the processing cycle. Incomplete removal can result in voids which have a detrimental effect on the mechanical properties of the composite;The introduction of secondary material processing operations such as fiber spinning which could increase the cost of the final product would be avoided [6].

### 2.3. Hybrid Woven Fabrics

Hybrid woven fabrics (HWFs) are characterized by orthogonal interlacing of two sets of yarns: polymer filaments and reinforcing long fibers in different architectures (carbon, glass, aramid, and natural fibers) constituting the fill (warp) and weft, respectively. There are many types of weaving patterns employed to create HWFs: plain weave, twill, and satin [44]. According to Figure 9 the woven fabric involves periodicity in its microstructure, thus permitting identification of a repeating unit, defined as unit cell. The index n_g_ can be defined to indicate the interlacing counts between the fill (polymer) and weft (reinforcement) yarns. The n_g_ index is equals two for plain weave, three for twill weave, and four or greater for satin weave [45].

Irrespective of the type of polymer matrix and reinforcing fibers, the weave pattern used for making the HWF contributes to the mechanical properties of the composite. For instance, research conducted by Alavudeen et al. [46] on natural fiber-reinforced hybrid polyester fabric highlighted that plain pattern exhibited higher tensile strength than the twill weave pattern due to a more uniform distribution of stress transfer with the application of tensile load in both the longitudinal and transverse directions. Indeed, in the plain weave configuration, fibers are interlaced one-to-one in both the warp and weft directions whereas there is no interlacing of filament in the weft direction in the twill pattern, which in turn reduces the strength properties of the composite. Fabrics are transformed by hot-pressing consolidation process into rigid composite structures of fibers embedded in a continuous polymer matrix. Using HWF in composite materials, the manufacturing process stay the same with respect to the film-stacking technique. Processing cycle consists of heating a stack of several layers of hybrid fabrics above the melting point of the matrix fibers, applying sufficient pressure to reduce the thickness of the laminate to the fully consolidated thickness and then cooling below the glass transition temperature of the matrix while maintaining pressure. The time to reach full consolidation depends on the time taken to fully impregnate the reinforcing fiber bundles. Processing cycles may be prolonged to improve the fiber impregnation and matrix-to-fiber adhesion or to achieve the desired level of crystallinity in the polymer matrix. Pressure, temperature, and holding time are the main process parameters governing the characteristics of the final composite [47].

Compared with the common unidirectional laminates, HWF composites provide better dimensional stability over a large temperature range, more balanced properties in the fabric plane and the interlacing of yarns results in higher out-of-plane strength, better impact resistance, and tolerance [48]. In the warp-weft configuration, after thermo-compression, the different filaments scatter amongst one another, as a result, the resin flow distance for impregnation can be greatly reduced and consequently the applied pressure and time are limited compared to film stacking and even considering the powder-based impregnation technique, leading to a saving in manufacturing costs [49]. However, intercalation of the polymer filament in the fiber bed by weaving would establish a fixed matrix-to-reinforcement proportion ratio in each hybrid fabric produced, limiting the management freedom in designing composites at varying reinforcement level.

On the best authors’ knowledge, recent literature regarding optimization of the manufacturing process of hybrid fabric as well as the effect of thermo-forming parameters on the final composite’s characteristics is very limited. Below, some topical studies are reviewed.

#### 2.3.1. CF-GF/Thermoplastic (PP/PET) HWFs: Weaving Process Optimization for Electromagnetic Shielding Applications

Lin et al. [50] combined PET and PP yarns and CFs and GFs to form woven fabrics by rotor twister method. Specifically, the authors investigating various rotary rate (from 9000 rpm to 15,000 rpm) during the manufacturing stage to optimize the structure of the fabric in terms of compaction between the wrapped yarns and tensile properties. The highest rotatory speed (15,000 rpm) yielded the slightest thickness of yarn and optimal tensile strength along the weft direction. In addition, the compact interior structure of hybrid fabric already achieved at 13,500 rpm enhanced the reflection and refraction of electromagnetic waves, conferring optimal shielding properties. Comparing GFs and CFs, the latter possessed higher electromagnetic effectiveness because of their greater electric conductivity that make the fibers able to shield the waves by absorbing them.

#### 2.3.2. GF/PP HWFs: Assessment of the Moulding Technology and Stacking Sequence

Formisano et al. [51] studied the influence of the manufacturing technology (isothermal and non-isothermal moulding methods) on the mechanical behavior of laminates from commercial hybrid fabrics of GF (60 *w*/*w*%) and PP fibers (40 *w*/*w*%) (Figure 10). The composite laminates (from 1 to 5 mm thickness) were realized with two stacking sequence, (1) 45–0–90 and (2) 0–90, to also evaluate the effect of laminate orientations on the mechanical characteristics. At the same moulding technology, 0–90 samples showed a higher peak force, penetration energy, and peak deformation values than 45–0–90 specimens. This result was attributed to several factors: higher bending stiffness and lower stiffness mismatch between adjacent layers in 0–90 configuration and greater sensitivity to delamination and interlaminar damage of 45–0–90 specimen. Considering the moulding methods, isothermal procedure was demonstrated better in term of strength, leading to a flexural strength almost 25% higher than those achieved by non-isothermal method.

#### 2.3.3. CF/PA6 HWF: Assessment of Impact Energy Absorption Capability

Di Benedetto et al. [52] developed a multiple regression model to predict the low-velocity impact behavior of CF/PA6 composite laminates obtained by compression moulding of five HWFs with stacking sequence 0°.

The purpose of this study was to optimize the energy absorption properties of composite and then increase their crashworthiness for using on automotive components. Five consolidation temperature (240 °C, 250 °C, 260 °C, 270 °C, and 280 °C) and two pressure conditions (<0.3 MPa and 0.3 MPa) were investigated to build the predictive model.

Processing temperatures above 260 °C negatively affected the dynamic mechanical response of the composite due to thermos-oxidative degradation phenomena that the polymer matrix underwent.

Indeed, in accordance with the thermal degradation kinetics method implemented by the authors, the material crashworthiness is highly associated with the degradation kinetics. The composite is more resistant when the degradation rates are reduced. With regard to the pressure, the laminates manufactured at 0.3 MPa provided a homogeneous and defects-free microstructure (Figure 11a) compared to the sample processed with pressure below 0.3 MPa (Figure 11b).

#### 2.3.4. Jute/PP HWF: Manufacturing and Characterization

Souza et al.’s research work [53] proposed the manufacture of Jute/PP (40 *v*/*v*% Jute–60 *v*/*v*% PP) HWFs as preforms for bio-composite laminates fabrication. The woven architecture and the cross-sectional optical micrographs of the laminate obtained by compression moulding (processing temperature of 190 °C, pressure of 0.2 MPa, and holding time of 20 min) are reported in Figure 12, respectively.

The HWF technique was successfully applied in the manufacturing of Jute/PP hybrid bio-composites by compression moulding. The obtained composites presented moderate mechanical properties of tensile strength (44.62 MPa) and elastic modulus (7.1 GPa), which were slightly better than the values found in literature on non-continuous fiber Jute/PP composites with same fiber/matrix proportion ratio. These performances suggest their possible application in non-structural components for reducing weight, costs, and environmental impact. The HWF has also presented interesting properties of malleability and strength required by textile industries for weaving processes.

### 2.4. Hybrid Yarns

The use of hybrid yarns is one of the most promising routes for producing structural thermoplastic composites due to a low-cost manufacturing of complex shaped parts. This process can offer a very good distribution of matrix and reinforcement in a non-molten state before processing. To take advantages of the processing and fabrication reductions that can be provided by textile technology, the thermoplastic matrix should be incorporated into the yarns before their conversion into preforms. In hybrid yarns, indeed, matrix and fabric are blended intimately at the filament level, such as the polymer flow distance for impregnation is reduced when matrix and fibers are commingled each other. To obtain an excellent and rapid impregnation, it is important that reinforcement and matrix are in direct contact as much as possible and distributed in a homogeneous way, reducing the distance that the molten matrix must take to wrap the reinforcement. [54]. To benefit from these advantages, it is necessary to take into account process parameters, identify and optimize them.

There are many methods by which the hybrid yarns can be manufactured such as co-wrapping and commingling. These techniques involve bringing the matrix and reinforcement into close contact with the aim not only to obtain, as already mentioned, a uniform distribution but also to limit the detrimental effects.

#### 2.4.1. Co-Wrapping Method

The yarns made by the co-wrapping method, called co-wrapped yarns, have a feature in which the reinforcing fibers are intimately combined with the thermoplastic matrix fibers using a self-assembled winding apparatus [55]. Referring to the process scheme presented by Zhai et al. [56], the co-wrapped yarns (CWYs) are fabricated by a hollow spindle spinning loom. Reinforcing fiber roving passes through the roving condenser and drafting rollers as a core roving then is guided into the hollow spindle; at the same time, matrix filament wraps around fiber roving then passes together in the hollow spindle. Reinforcing fiber roving and polymeric filament both have false twist since the hollow spindle rotated speedily. After passing through the twisting hook, the false twist of reinforcing fiber roving become untwisted, while matrix filament wraps remain twisted (Figure 13). A crucial parameter for co-wrapping method is the cover factor (CF), which represents the percentage of fiber roving surface covered by polymer filaments and the related to the proportion ratio between matrix and reinforcement [56]. CF can be estimated as follows (Equation (1)):(1)$$\mathrm{CF}=1-{\left(1-\frac{W_{M}\times T_{M}}{1000}\right)}^{2}$$
where W_M_ is the polymer filament width (mm) and T_M_ is the polymer wrapping turns per meter (turns/m).

With respect to other hybrid yarn production methods, such as commingling process, co-wrapping ensures better protection for the reinforcing fibers during manufacturing. However, the distribution of the reinforcing fibers and the matrix fibers is still a challenge, requiring higher processing temperature and pressure to improve impregnation [57]. As for commingled, in co-wrapped fabric the impregnation takes place mainly by resin flow perpendicular to the fibers so the impregnation time are generally longer than powder-based process. In addition, one of the main advantages of co-wrapped fabrics is their freedom design. Hybrid yarns are drapable to a variety of complex and their dimensional shaped parts avoiding further forming/moulding processes [58]. A recent literature survey on co-wrapping technology revealed a lot of research work investigating both the synthesis parameters to produce the hybrid yarn and the optimization of compression molding processing to achieve the composite laminates. Mirdehghan et al. [59] investigated the influence of linear densities (16, 50.67, and 67.11 tex) and wrapping densities (115, 180, and 230 turns/m) of polyester fibers on the tensile performance of glass/polyester co-wrapped hybrid yarns. The authors found that the breaking load and tenacity of yarn increased with an increase in the wrapping density until an optimum point (180 turns/m). After that, increment in wrapping density produced a decrease in mechanical properties. Excessive twisting of glass fibers would lead to morphological alteration and damage, negatively affecting the tensile strength.

In addition, the intermediate value of linear density (50.67) was discovered as optimal in terms of mechanical properties, providing the best condition for the radial compressive force exerted by the wrapped filament to interact with glass core. At the optimal process conditions, co-wrapped yarns provided breaking load and tenacity 62% and 46% higher than those of the neat glass fibers (not wrapped with polyester). Baghaei et al. [60] studied PLA-hemp co-wrapped hybrid yarns evaluating the effect of wrap density (150 wraps/m and 250 wraps/m), and PLA/hemp mass ratios (10/90, 20/80, 35/65, and 45/55) on the hot compacted laminate composites. Tensile test results (Figure 14) revealed that composites with a wrapping density of 250 showed a significant improvement in tensile modulus over the composites fabricated at 150 wraps/m, across the whole range of fiber fraction ratios investigated. At a low reinforcement level (<25% by mass), the improvement was approximately 4.6%; at higher fiber fraction (>40% by mass), the improvement increased to approximately 14.7%. This increase could be attributed to the better fiber alignment. Increasing wrapping density improved the tensile modulus of the composites because the tortuosity of the hybrid yarns decreases as wrap twist increases, and consequently the reinforcement alignment would be improved. Xu et al. conducted a comparative study between PEEK-based co-wrapped yarns using carbon [57] and glass fibers [55] as reinforcement materials. Specifically, the authors researched the optimal laminate manufacturing process investigating the following process parameters: molding temperature (370–450 °C), holding time (10–150 min), and cooling rate (from −2 °C/min, −10 °C/min, and −20 °C/min). The best results of the above-mentioned works are reported in Table 2. In both cases, intermediate values of the process parameters provided the maximum mechanical performance. Temperatures and holding times higher than the optimal value were deleterious for the mechanical behavior of the composites due to the thermal degradation that the PEEK matrix underwent. A cooling rate of 10 °C/min was detected as an optimum condition in terms of crystallinity degree achieved in the examined matrix [55,57]. It is worth mentioning that the carbon-based laminates were processed at lower temperatures and holding times than the glass-based counterpart.

This is to be attributed to the higher thermal conductivity of carbon fiber (~1.31 W/m × K) compared to glass (~0.25 W/m × K) which would sanction a more effective heat transfer at the fiber–matrix interface and therefore not requiring higher temperatures to drive the melting of the matrix and the consequent impregnation of the reinforcement [61].

#### 2.4.2. Commingling

Commingling is one of the most promising techniques founded on the principle of uniform distribution of continuous matrix and reinforcement filaments during melt spinning [62]. The homogeneous fiber/matrix distribution of commingled yarns in particular leads to short impregnation routes and low void contents reflected by the high mechanical performance of the thermoplastic composites [63]. This method allows for acceptable levels of impregnation to be attained, but relatively long cycles are required. In the follow Figure 15 [64] is shown the yarn cross section. Another important advantage is that commingled yarns can be processed and produced entirely with almost all existing known technologies for the fabrics manufacture. Combined with the developments in textile structures, the use of commingled yarns considerably improves the mechanical properties of the resulting composite parts [65]. As Figure 16 [66] shows, commingling process can be schematized as follows: the multifilament yarns cross through a section on which a jet of compressed air acts, so that tangles are created inside them. The result at the section is the intermingled yarn, obtained by processing several yarns from which the single thread is obtained. The Figure 16, precisely, draws the intermilling process of a polymer-metal yarn carried out by Özkan et al. [66] for composites intended for electromagnetic applications.

Due to the commingling process, both reinforcing and polymer fibers can be combined, representing great potential [67,68]. The desired ratio of fiber to matrix can be achieved by varying the number of constituent yarns during the production of the hybrid yarn itself.

Nevertheless, the process involves technological challenges concerning the impregnation as well as wetting of the fibers hence a decrease in the quality of the consolidation [69]. The atmospheric pressure, production speed, machine configuration and designing of the nozzle, filament stiffness, fiber cross-sectional area and the ratio of the diameter [69], density of the matrix forming fibers, the cross-section of matrix forming filaments, the number of filaments in yarn bundle, filament diameter, diameter ratio of reinforcing and matrix forming filaments influence the degree of commingling and structural attributes of hybrid yarns [20]. The mixing quality of commingled yarn increases and decreases directly with the rate of overfeeding and processing speed, respectively, which is given by
(2)$$D_{\mathrm{ov}}=\left(\frac{V_{\mathrm{DG}}-V_{\mathrm{TG}}}{V_{\mathrm{TG}}}\right)\times 100{\;\%}^{}$$
where V_DG_ and V_TG_ represents, respectively, the speed of delivery godet and the take-off godet [70]. Numerous studies have concentrated on commingled materials, yarn manufacturing and preforming, and mechanical properties of commingled composites [71,72]. Jumaev et al. [73] reported a wide range of experimental and analytical investigations into parameters that affect the performance of commingled composites, such as damping properties, hardness, coefficient of friction, moisture absorption, and other mechanical properties. However, by employing a highly dissipative epoxy resin, it is feasible to obtain a high flexural modulus and a loss factor in the case of carbon-reinforced unidirectional plastic composite. Conversely, maximum energy dissipation occurs near the glass transition temperature, which also significantly reduces the other mechanical properties. Mankodi et al. [74] examined the impact of commingling parameters on PP/GF composites properties. According to their study, the linear density value of the resulting composite is significantly affected by overfeeding and air pressure, which in turn results in poor quality and high material failure. The overfeed value also affects the toughness of PP/GF hybrid yarns. At low take-up, speed tenacity improves as a function of pressure as well as overfeed, whereas the higher winding speed leads to poor yarn tenacity with reduced nip frequency and poor regularity. Di Benedetto et al. [52] reported energy dissipation from low velocity impact test considering processing parameters, material properties and degradation kinetic of the matrix of polyamide/carbon fiber commingled composite, that is the capability to absorb energy and then increase crashworthiness. In this study it is seen that the energy absorption capability must be increased to promote thermoplastics materials composites reliability for using on automotive components. The viscosity of the PA6 multifilament yarn undergoes a significant decrease after the temperature of 220 °C, due probably to the molecular mobility caused by the polymer melting. In this case, intermolecular forces of attraction were not sufficiently strong to resist the forces induced by the rheometer. The increase of the temperature up to 220 °C increased the vibrational motion in the polymer molecules causing them to lose their intermolecular forces of attraction and undergo easy movement. The increase of the processing temperature causes a reduction of the resistance force and total energy of the material. A processing temperature above 260 °C affects dissipated energy values due to the matrix degradation. Lu et al. [75] analyzed four parameters for hot-compacted CF/PEEK plain weave fabric composites: melting temperature, molding pressure, crystallization temperature and the mass content of resin. It is seen that a higher compaction temperature leads to more matrix creation and better layer bonding; indeed, a suitable molding pressure causes bubbles to be discharged from composites and helps to form a uniform surface, but when the pressure becomes too high, this is expected to result in loss of resin content, resin outflow, and probable distortion of fiber bundles. Moreover, the melting temperature has a non-negligible effect on the static contact angle of PEEK on CF mats and it means that the smaller the contact angle, the better the wettability. Samples crystalized at 300 °C show excellent tensile properties and crystallinity. Resin mass content is also an important parameter in the fabrication of composites: The high mass ratio of PEEK material provides good interfacial strength, whereas the increased resin content causes loss of resin, dispersion of carbon fiber, and creates defects while processing. Lekube et al. [76] investigated the influence of porosity, fiber content and fiber length as well as processing parameters on the properties of partially compacted nonwoven composites based on polypropylene and glass fibers. The study showed an increasing density with increasing glass fiber content. Moreover, stiffness and strength of the composites increase as well when glass fiber content increases. In addition, both the elastic modulus and tensile strength increase with increasing initial glass fiber length. Furthermore, important factors for restriction of void content are the number of stacked layers and applied pressure during processing. The study of Bernet et al. [77,78] has foreseen the numerical simulation of the impregnation and consolidation process in order to obtain composite materials made with hybrid yarns. The developed model involved the use of both carbon fibers reinforcement and PA12 polymer fibers matrix. At the beginning of the impregnation it is assumed that the commingled yarn is like a molten resin surrounding the reinforcing fibers [79], as shown in Figure 17 [63]. The consolidation is regulated by the impregnation of the fiber agglomerations by the resin, at a rate which is presumed to follow Darcy’s law [20]. Specifically, the following figure illustrates the scheme processing of online commingled yarns (thermoplastics + glass fibers) proposed by Wiegand and Mäder [63]. In their work, the authors investigated different matrices (PP, PLA, and PA) and sizing chemistry on the mechanical performance and consolidation behavior of glass fiber-based commingled yarns. For all examined matrices, significant improvements in fiber/matrix compatibility were detected by using pure silane coupling agents as sizing component.

Compared to the production processes analyzed so far, the hybrid yarns technique is certainly the most ambitious and promising since they have some non-negligible advantages such as high flexibility, and this allows to obtain more complex parts [20]. Commingled composite is an innovative and inventive systems with research from industries and academia still at work to study and fabricate new commingled composite, characterize them for perfect function, making them exemplary candidates, particularly in structural applications. However, in the enduring literature, no coalescence of comprehensive details is available to the author’s foremost information and understanding.

## 3. Key Characteristics, Pros and Cons of Investigated Techniques

Thermoplastic composites can play a key role in 21st century industry, as new materials are emerging day by day, while manufacturing processes are evolving to meet the stringent industrial performance requirements, production and multifunctionality. This section aims to summarize and highlight the key aspects of each technique, the various advantages, disadvantages and the parameters that most influence and drive processes choices that cannot be ignored. Film stacking molding is widely used due to the relative ease of manufacture. Disadvantages include high resin content, the uneconomic nature of the process, and the difficulty in impregnating the fiber tow (high pressure forces the fibers together) with high-viscosity matrix material. We have seen how important processing temperature, pressure and holding time are, as the core parameters to control the process. The higher the temperature, the lower the resin viscosity, an advantage that guarantees better mobility and fluidity through the fiber bed. On the other hand, however, high temperatures can degrade the material, nullifying its contribution. Pressure represents a crucial parameter: if too low, the impregnation time increases with the consequence to obtain dry spot inside the composite. Nevertheless, an excessive increase would cause a reduction in the fabric permeability. In the powder impregnation process sufficient pressure and time must be applied to force the resin to impregnate the fiber tows in order to achieve fully consolidated composite parts. Poor impregnation and consolidation will result in a significant reduction in the mechanical properties of the final composites [80]. We report some of the most important factors influencing the process: particle size, fiber tow spreading, particulate flow pattern. Regarding hybrid woven fabrics, composites manufactured with this technique offer an effective way of increasing ultimate strain and impact properties while reducing the cost of an advanced composite material. By combining two or more types of fibers, it is possible to obtain the advantages of both the fibers while simultaneously mitigating their fewer desirable qualities. The mechanical properties of a hybrid composite can be varied by changing composite volume fraction and stacking sequence of different plies. Fibers such as carbon or boron are widely used in many aerospace applications because of their high specific modulus [80]. Hybrid yarns, consisting of reinforcing and matrix fibers, are one kind of basic material (semi-finished product) with which to construct continuous-fiber-reinforced thermoplastic composites [81,82]. Composite properties are influenced mainly by the arrangement of the reinforcing fibers and the homogeneity of the fiber’s distribution in the composite, as well as by impregnation of the glass fibers with the polymer matrix. Hybrid yarns are usually manufactured into thermoplastic composites by hand lay-up [83], filament winding [84] or—as done recently—by the pultrusion process [85]. The incredible advantage of this technique, which distinguishes it from the others and makes it the most prestigious is the possibility to low-cost manufacture complex shaped parts, owing to reduced impregnation times and applied pressures during processing.

An aspect of extreme importance lies in the fact that these are industrial production processes and that allows to start from thermoplastics obtained from recycled material.

## 4. Conclusions

This review article investigated the main production technologies to produce thermoplastic matrix composite materials. The reason lies in a principle that is now more important than ever, sustainability: the opposite nature of thermoplastic matrix composites compared to thermosets, i.e., the possibility to recycle them. The thermosetting polymer certainly guarantees the best mechanical characteristics but it can no longer be reworked and reused for other purposes. Conversely, thermoplastic polymer can be remelted and subjected to a new type of processing. The goal is to produce high-performance thermoplastic matrix composite materials, with high volume-based quantities of reinforcement, to allow a real comparison with thermosetting materials. There are two process techniques to manufacture thermoplastic composite parts: reactive and melt processing. The first solution focuses on in-situ polymerization, starting from water-like viscosity monomers that could potentially provide an optimal reinforcement impregnation, and a subsequent polymerization reaction triggered by temperature increase [7]. The second one lies in melt processing, discussed in detail in this article. Four main technologies have been revised such as film stacking molding, powder impregnation, hybrid fabrics, hybrid yarns and the respective parameters that most influence the processes. Hybrid yarn is certainly the most technologically advanced but hard process. It opened a new era by kicking off the research area with their unique advantage of easy production and prodigious application. Commingling is the most favorable method for several industrial applications owing to their process simplicity and cost-effectiveness, which is imperative for combining its advantage with other conventional technologies [69]. Numerous combinations of materials and reinforcements are now available on a wide range of length, innovative material shapes and different processing possibilities for both designers and fabricators.

## Figures and Tables

**Figure 1 polymers-15-00242-f001:**
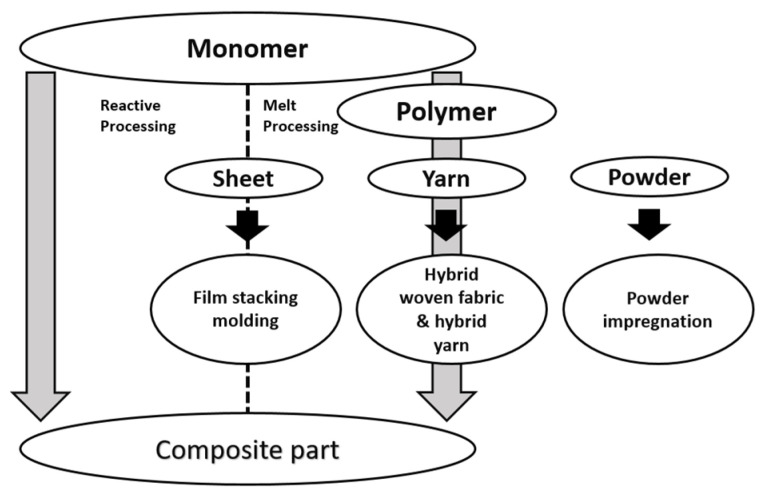
Processing steps for manufacturing thermoplastic composite parts through melt and reactive processing. Author’s own figure.

**Figure 2 polymers-15-00242-f002:**
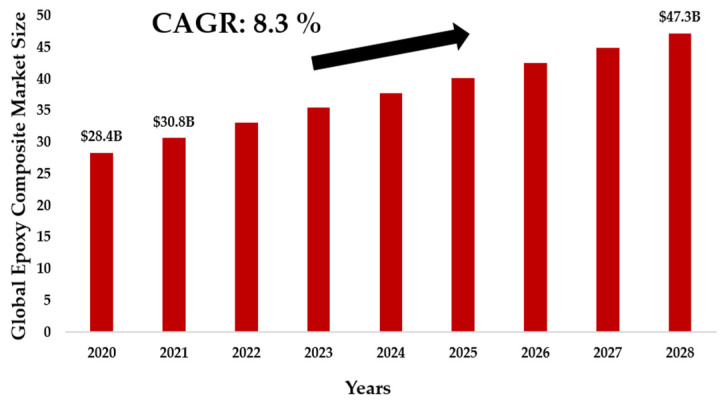
Epoxy Composites Market Size and Forecast. Authors’ own figure from Reference [19].

**Figure 3 polymers-15-00242-f003:**
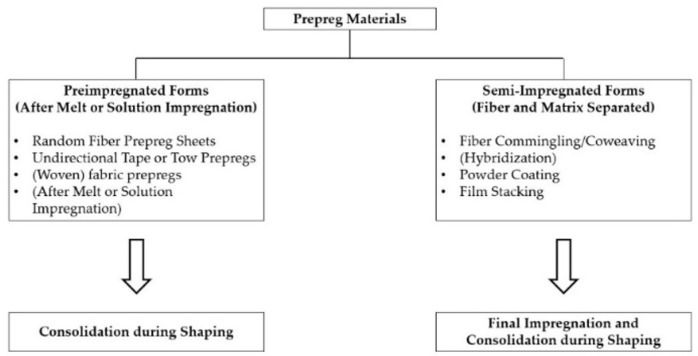
Typical impregnation techniques for thermoplastic composites materials. Authors’ own figure from Reference [6].

**Figure 4 polymers-15-00242-f004:**
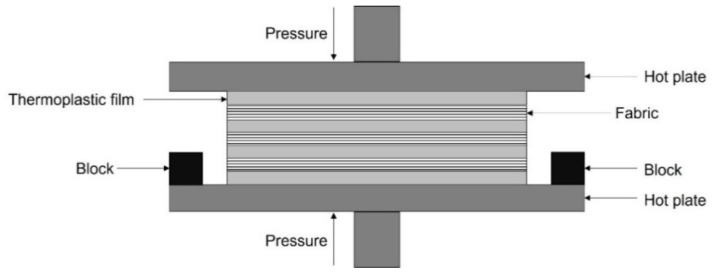
Schematic of film stacking process for thermoplastic composite laminates. Author’s own figure.

**Figure 5 polymers-15-00242-f005:**
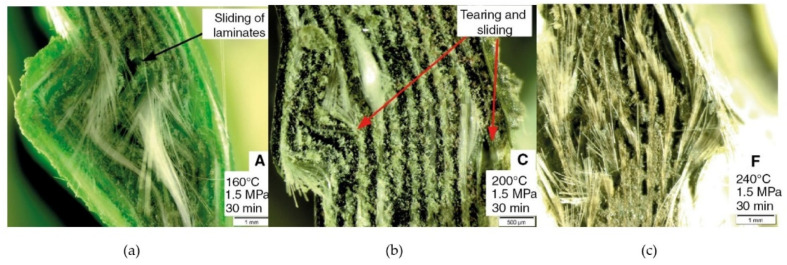
Failure mechanism in GF-PVC laminate as a function of processing temperature: (**a**) 160 °C, (**b**) 200 °C, (**c**) 240 °C. Adapted with permission from Reference [27].

**Figure 6 polymers-15-00242-f006:**
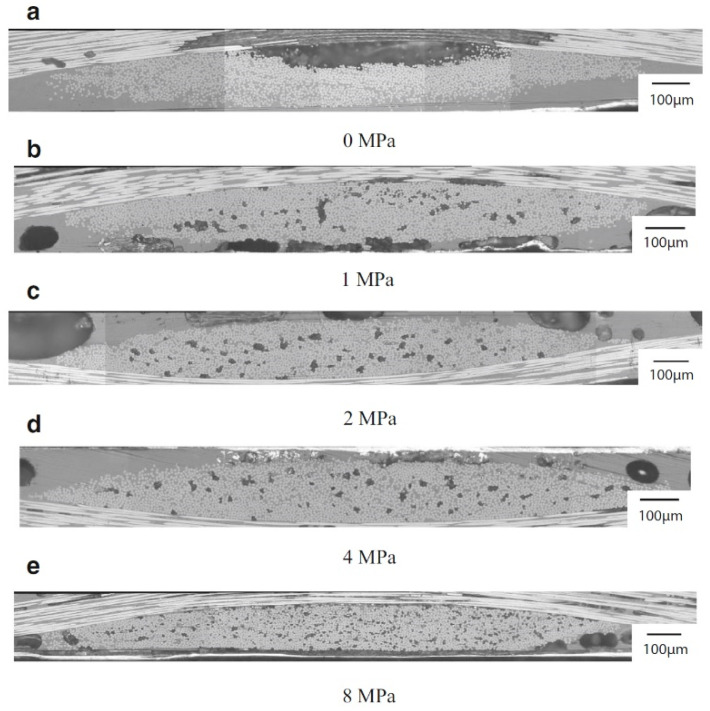
Cross section of CFF-PI laminate as a function of mould pressure: (**a**) 0 MPa, (**b**) 1 MPa, (**c**) 2 MPa, (**d**) 4 MPa, and (**e**) 8 MPa. Reprinted with permission from Reference [32].

**Figure 7 polymers-15-00242-f007:**
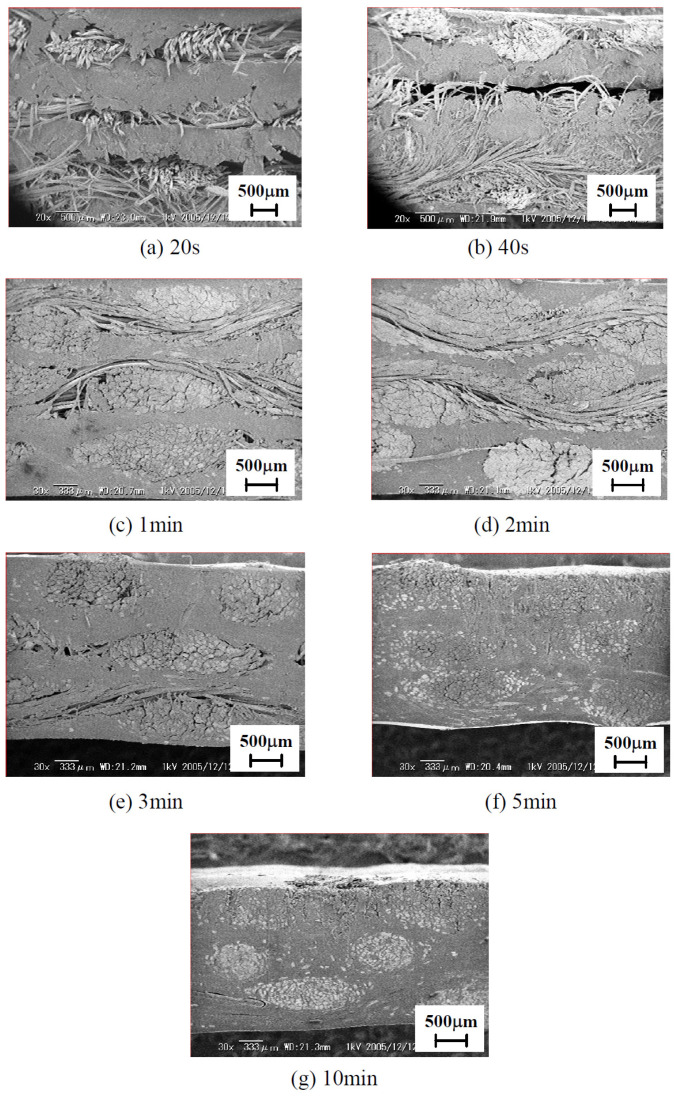
Cross section of JF-PLA laminate as a function of holding time: (**a**) 20 s, (**b**) 40 s, (**c**) 1 min, (**d**) 2 min, and (**e**) 3 min, (**f**) 5 min, and (**g**) 10 min. Reprinted from Reference [31] under Creative Commons license.

**Figure 8 polymers-15-00242-f008:**
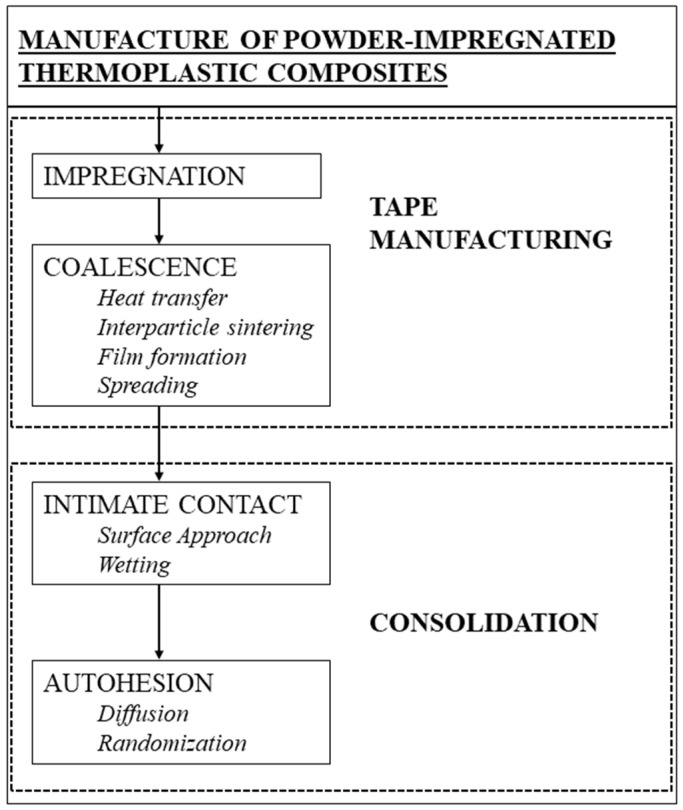
Flowchart for the manufacture of powder-impregnated thermoplastic composites. Authors’ own figure from Reference [6].

**Figure 9 polymers-15-00242-f009:**
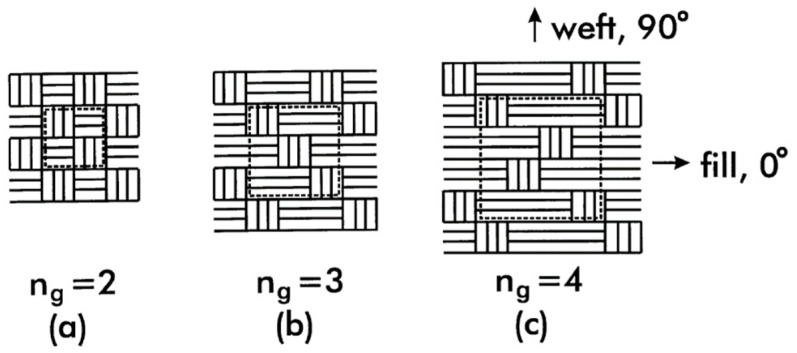
Different weaving patterns in hybrid woven fabrics: (**a**) plain, (**b**) twill, and (**c**) stain. Reprinted with permission from Reference [45].

**Figure 10 polymers-15-00242-f010:**
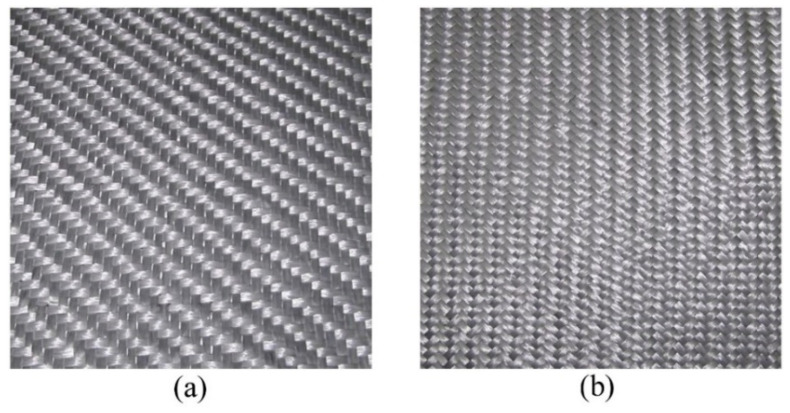
PP-GF fabrics investigated by Formisano et al.’s research [8]: (**a**) 0/90 ply and (**b**) ±45 ply. Adapted with permission from Reference [51].

**Figure 11 polymers-15-00242-f011:**
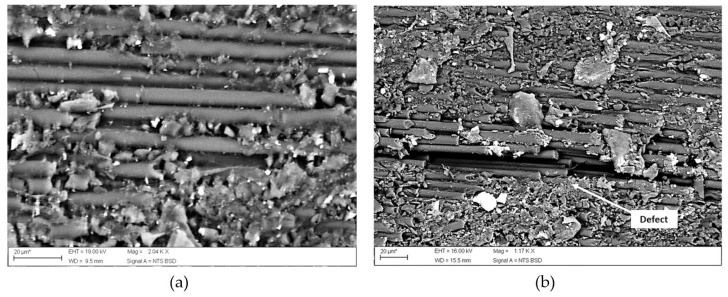
CF/PA6 laminates processed with pressure (**a**) of 0.3 MPa and (**b**) below 0.3 MPa. Adapted with permission from Reference [52].

**Figure 12 polymers-15-00242-f012:**
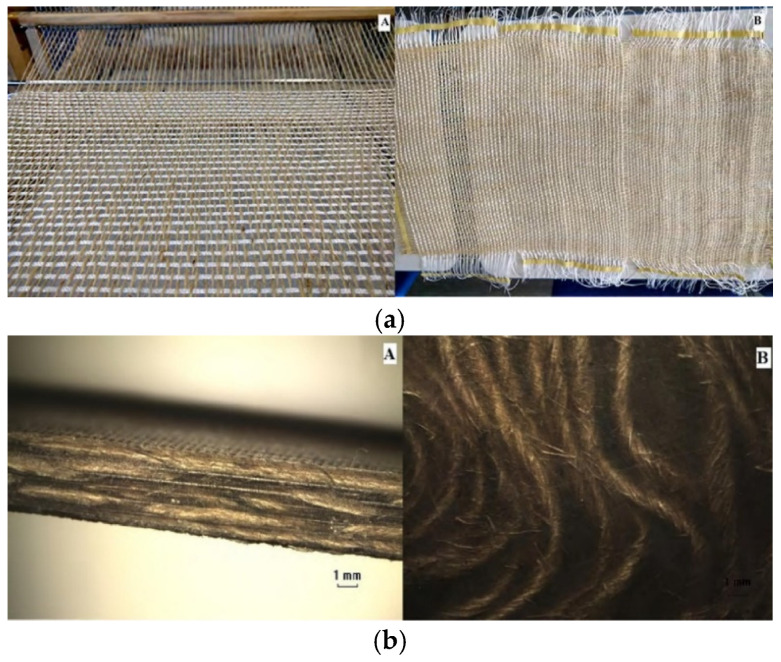
(**a**) Jute/PP HWF architecture and (**b**) cross-sectional optical micrographs of the laminate Adapted from Reference [53] under Creative Commons license.

**Figure 13 polymers-15-00242-f013:**
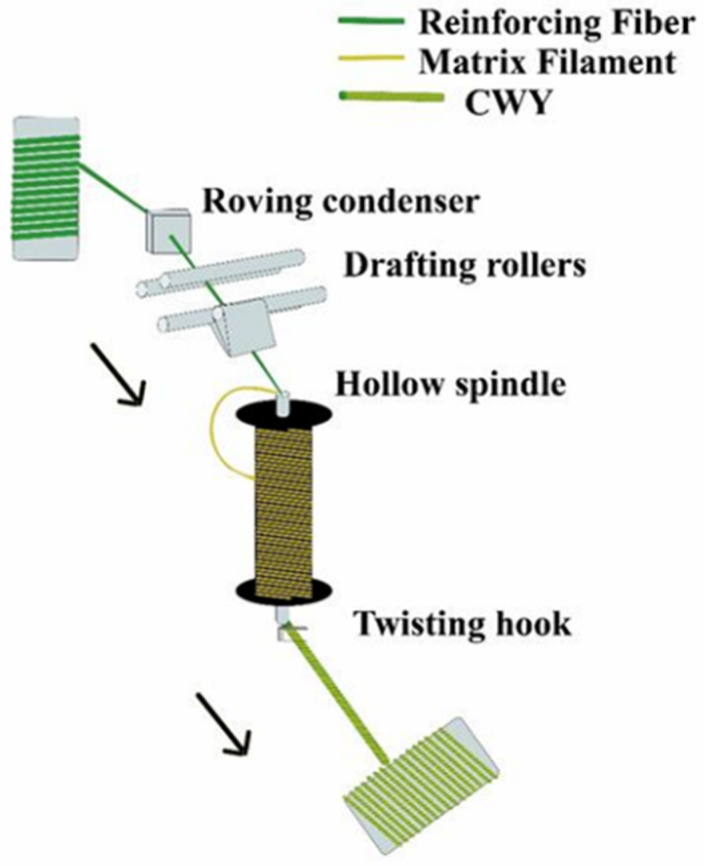
Schematic of co-wrapping process for hybris yarns. Adapted from Reference [56] under Creative Commons license.

**Figure 14 polymers-15-00242-f014:**
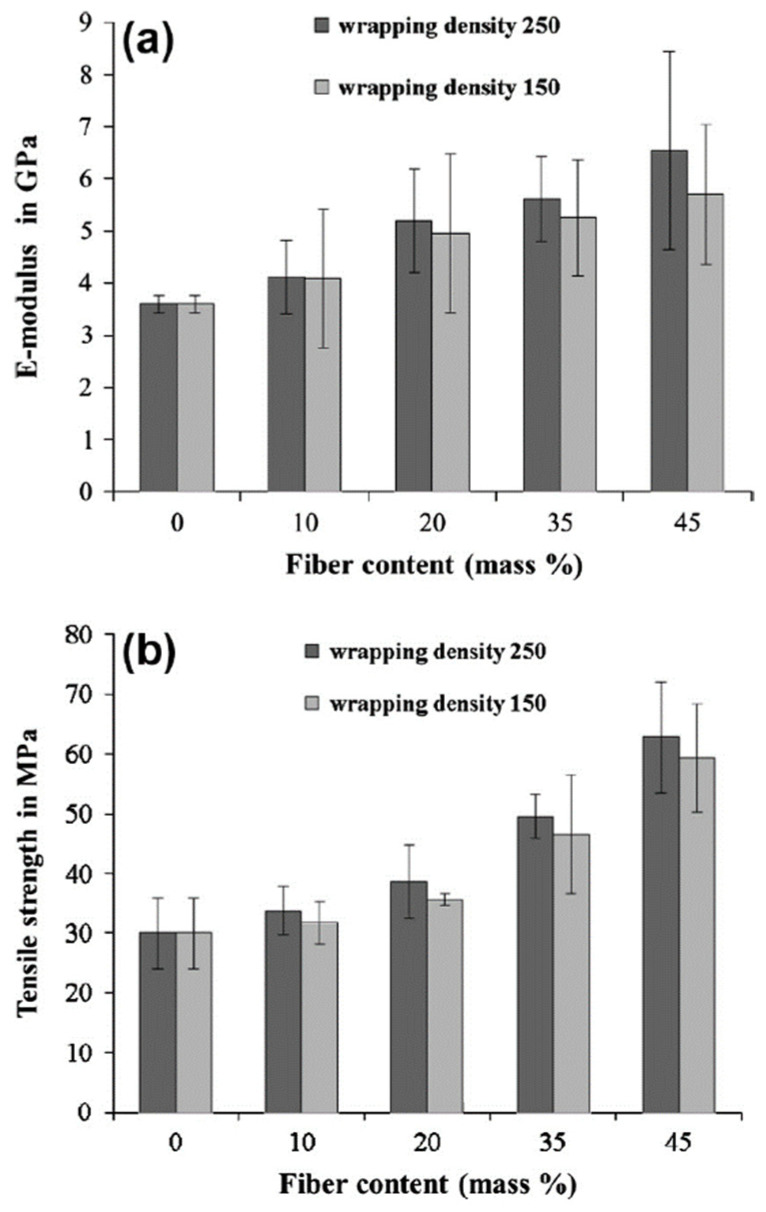
Mechanical properties in tensile of co-wrapped PLA-hemp laminates: (**a**) elastic modulus and (**b**) tensile strength. Adapted with permission from Reference [60].

**Figure 15 polymers-15-00242-f015:**
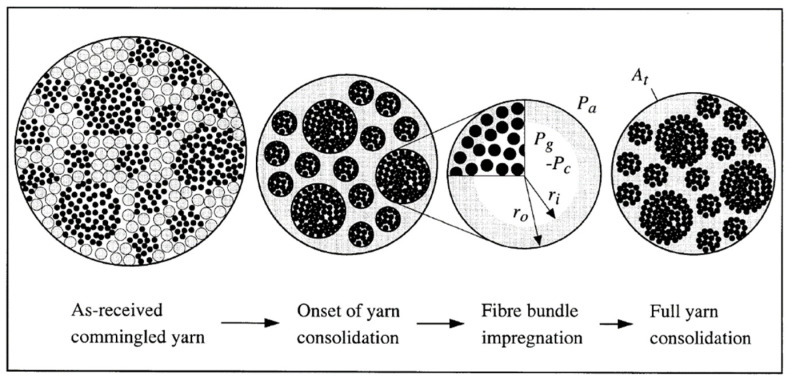
Representative commingled yarn cross-section and assumed consolidation mechanism. Reprinted with permission from Reference [64].

**Figure 16 polymers-15-00242-f016:**
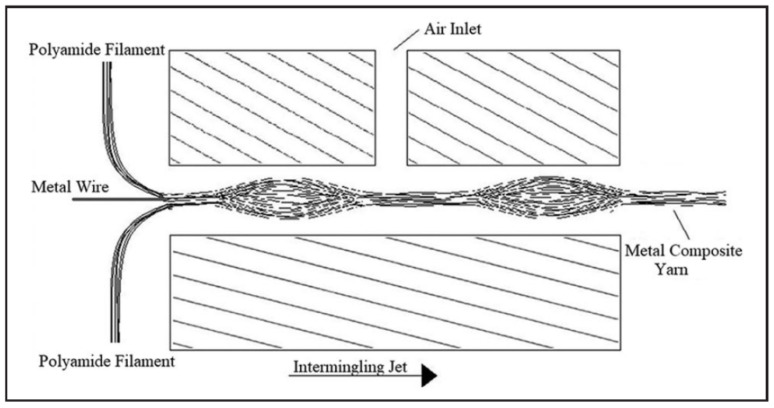
Intermingling process. Reprinted from Reference [66] under Creative Commons license.

**Figure 17 polymers-15-00242-f017:**
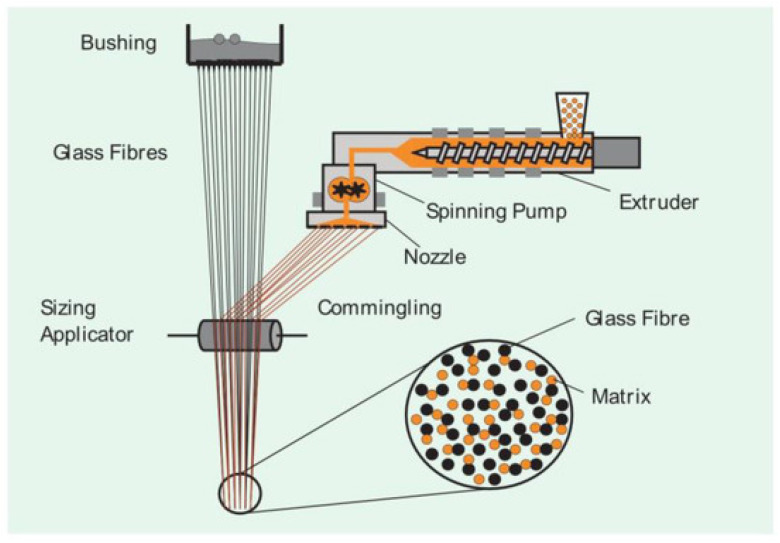
Processing scheme for commingled yarn. Reprinted from Reference [63] under Creative Commons license.

**Table 1 polymers-15-00242-t001:** Reference values of thermoplastic polymers. Authors’ own table from [20].

Materials (Thermoplastic Polymer)	Tensile Modulus, GPa	Tensile Strength (Yield), MPa	Melt Flow, g/10 min	Melting Point, °C	Density, g/cm^−3^
Polypropylene (PP)	1.50–1.75	28–39	0.47–350	134–165	0.89–0.91
Polyethylene (PE)	0.15	10–18	0.25–2.6	104–113	0.918–0.919
Polyamide (PA)	0.7–3.3	40–86	15–75	211–265	1.03–1.16
Poly ether ether ketone (PEEK)	3.1–8.3	90–11	4–49.5	340–344	1.3–1.44

**Table 2 polymers-15-00242-t002:** Optimal process parameters and performance.

Laminate	Temperature	Time	Cooling Rate	Tensile Strength	Tensile Modulus
PEEK-glass [55]	430 °C	120 min	−10 °C/min	1.24 GPa	29.62 GPa
PEEK-carbon [57]	400 °C	60 min	−10 °C/min	1.51 GPa	161 GPa

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
