# Peer review of "Different Production Processes for Thermoplastic Composite Materials: Sustainability versus Mechanical Properties and Processes Parameter"

_polymers, 2023, doi:10.3390/polym15010242_

Round 1

Author Response

In Answers to Rev-1 there are all requested notices

Author Response

All is reported in Rev-2 answers

Round 2

Reviewer 1 Report

I see that the authors have improved their work.

I still believe the abstract could use stronger language to summarize the key methods reviewed. Not just write at the end methods such as are reviewed.

Figure 2 is now Figure 3, but in the text, you write, "Figure 2 presents several methods for wetting the fibers with a thermoplastic polymer." Fix it..

I do think you could have done a bit better job with Figure 1.

Apart from clearly promoting their article with key takeaways in the abstract and, possibly, title, I do think the review is interesting and fit for publication after English language and typographical corrections.

I would suggest asking for an extension of revision time next time, so it does not feel rushed.

Author Response

       Reply  to Reviewer 1:

  1. I still believe the abstract could use stronger language to summarize the key methods reviewed. Not just write at the end methods such as are reviewed.

The comment has been implemented has required.

  1. Figure 2 is now Figure 3, but in the text, you write, "Figure 2 presents several methods for wetting the fibers with a thermoplastic polymer." Fix it.

The error has been fixed.

  1. I do think you could have done a bit better job with Figure 1.

The revision times has been very close. In any case, we will take  your suggestion into account, especially because we are working on the next experimental work that we would like to present as a scientific study on reactive polymerization for the composite materials’ manufacture.

  1. A part from clearly promoting their article with key takeaways in the abstract and, possibly, title, I do think the review is interesting and fit for publication after English language and typographical corrections.

Thank you very much for your suggestion, we will certainly take it into account.

  1. I would suggest asking for an extension of revision time next time, so it does not feel rushed.

We agree with this remark.